# SciCode: A Research Coding Benchmark Curated by Scientists

**Minyang Tian**[1,2*‡], **Luyu Gao**[3*], **Shizhuo Dylan Zhang**[1], **Xinan Chen**[1†], **Cunwei Fan**[1†],
**Xuefei Guo**[1†], **Roland Haas**[1†], **Pan Ji**[4†], **Kittithat Krongchon**[1†], **Yao Li**[1†],
**Shengyan Liu**[1†], **Di Luo**[5,6,11†], **Yutao Ma**[7†], **Hao Tong**[1†], **Kha Trinh**[7†], **Chenyu Tian**[8†],
**Zihan Wang**[1†], **Bohao Wu**[1†], **Yanyu Xiong**[9†], **Shengzhu Yin**[1†], **Minhui Zhu**[1†],
**Kilian Lieret**[10], **Yanxin Lu**[1], **Genglin Liu**[1], **Yufeng Du**[1], **Tianhua Tao**[1],
**Ofir Press**[10], **Jamie Callan**[3], **Eliu Huerta**[1,2,7‡], **Hao Peng**[1‡]

[1]University of Illinois Urbana-Champaign [2]Argonne National Laboratory
[3]Carnegie Mellon University [4]University of North Carolina at Chapel Hill
[5]Massachusetts Institute of Technology [6]Harvard University [7]University of Chicago
[8]University of Texas at Austin [9]Stanford University [10]Princeton University
[11]The NSF AI Institute for Artificial Intelligence and Fundamental Interactions
* Equal contribution lead authors. † Data curation, alphabetical order.
‡ Correspondence to {mtian8, haopeng}@illinois.edu, elihu@anl.gov

### Abstract

Since language models (LMs) now outperform average humans on many challenging tasks, it is becoming increasingly difficult to develop challenging, high-quality, and realistic evaluations. We address this by examining LM capabilities to generate code for solving real scientific research problems. Incorporating input from scientists and AI researchers in 16 diverse natural science sub-fields, including mathematics, physics, chemistry, biology, and materials science, we create a scientist-curated coding benchmark, **SciCode**. The problems naturally factorize into multiple subproblems, each involving knowledge recall, reasoning, and code synthesis. In total, SciCode contains 338 subproblems decomposed from 80 challenging main problems, and it offers optional descriptions specifying useful scientific background information and scientist-annotated gold-standard solutions and test cases for evaluation. OpenAI o1-preview, the best-performing model among those tested, can solve only 7.7% of the problems in the most realistic setting. We believe that SciCode demonstrates both contemporary LMs' progress towards realizing helpful scientific assistants and sheds light on the building and evaluation of scientific AI in the future. [1]

## 1 Introduction

The development of evaluations in tandem with language models (LMs) has substantially contributed to the rapid advancement of these models [30, 12, 8, 26, 83, 28, 74]. Because LMs now surpass the performance of most humans except domain experts, evaluating them becomes increasingly challenging. Many established benchmarks struggle to keep pace with the advancements in LM performance and have quickly become saturated [93, 15, 72, 59], leading to discrepancies between the models' perceived and actual capabilities [37]. As a consequence, researchers are developing synthetic challenging benchmarks, often involving models in the construction of evaluation instances. For example, some subsample instances from existing benchmarks that cannot be solved by current models [95, 84], or augment them to construct more challenging evaluations [22, 45, 50]. However, it is unclear whether such efforts accurately reflect real-world applications and the models' performance in practical scenarios. Realistic, high-quality, and challenging evaluations are crucial for the continued advancement of LMs.

---

[1]Data, code, and leaderboard available at `https://scicode-bench.github.io/`

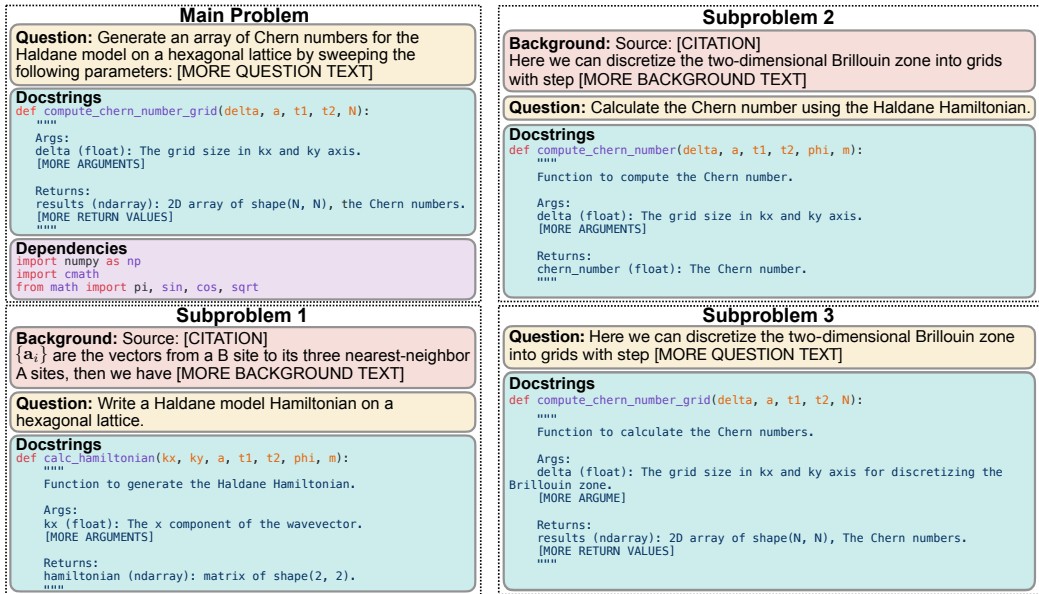

Figure 1: A SciCode main problem is decomposed into multiple smaller and easier subproblems. Docstrings specify the requirements and input-output formats. When necessary, scientific background knowledge is provided, written by our scientist annotators. The full problem is shown in subsection A.3

We therefore propose SciCode, a benchmark containing code generation problems drawn from diverse natural science fields, including mathematics, physics, chemistry, biology, and materials science. SciCode contains 80 main problems, each decomposed into multiple subproblems, totaling 338. Each problem provides the scientific background when necessary as well as detailed instructions. To solve it, the model must implement multiple Python functions—one for each subproblem—and then integrate them into a complete solution for the main problem. For every main problem and subproblem, SciCode provides gold-standard solutions and multiple test cases, facilitating easy and reliable automatic evaluation. Figure 1 shows an example.

SciCode aims to overcome the challenges of current LM evaluations by introducing the following value-added design choices.

- *Intentional focus on natural science fields*, such as computational mechanics, quantum information and computing, quantum chemistry, ecology, and molecular modeling.
- *Abundant high-quality data not usually made available to current LMs* [94, 1, 14], enabling a more robust evaluation of the models' ability to generalize to less familiar scenarios.
- *High annotation quality*, with all problems, including gold solutions and test cases, annotated, revised, and verified by at least two senior researchers (PhD student level or above) in represented scientific domains.
- *Realistic and current problems* sourced from scientists' everyday research tasks or influential papers. This ensures SciCode's relevance to real-world applications.
- *Problems curated to have zero overlap with publicly available datasets* to prevent potential data contamination.[2]
- *Problems that test LM's comprehensive and all-around capabilities*. Solving the main problems requires deep scientific background knowledge, strong analytical capabilities to decompose complex problems into simpler ones and correctly solve each, and the ability to integrate partial into complete solutions.
- *Opportunities to evaluate various model capabilities in varied setups* by toggling options, e.g., whether to provide scientific background information or to condition on gold or generated solutions to previous subproblems.

---

[2]In addition to addressing data contamination, we find that most problems are too challenging for even the best models. Therefore, we often simplify problem settings and provide more background during revisions.

Further, we believe that the availability of this well-designed benchmark can *motivate research into developing new AI methods for accelerating scientific research*, an area that has thus far benefited less from recent LM advancements partly due to a lack of commercial incentive.

We use SciCode to evaluate state-of-the-art proprietary and open models. Results show that SciCode is a very challenging benchmark: in the most realist evaluation setup, Claude3.5-Sonnet, the best-performing model in our experiments, can solve only 4.6% of the main problems, while other strong models, such as Claude3-Opus and GPT-4o, solve only 1.5%. Similarly, the best open source model under test, Deepseek-Coder-v2, can only solve 3.1% of the problems. The other open-source LLMs under test (e.g., Llama-3-70B-Instruct and Mixtral-8x22B-Inst) fail to complete any problems despite successfully solving some subproblems correctly. Our analysis finds that all models can benefit from the background knowledge written by our scientist annotators, achieving substantial and consistent improvements. However, even with background, the best model can solve only 12.3% of the main problems.

## 2 SciCode

This section examines the design principles and annotation process we chose for SciCode, describing: research-level coding problems from various natural science fields (§2.1); how we decomposed main problems into multiple, simpler subproblems (§2.2); our design choices for the annotation process (§2.3); and various evaluation setups that SciCode facilitates (§2.4).

### 2.1 Challenging and Realistic Scientific Coding Problems

SciCode sources challenging and realistic research-level coding problems across natural science disciplines, including mathematics, physics, chemistry, biology, and material science, covering a total of 16 subfields. This diverse selection ensures a comprehensive representation of the natural sciences, where extensive code development is essential.

SciCode is mainly drawn from the scripts that scientists use in their everyday workflow. Many of these have been used in one or more publications, demonstrating their robustness and correctness. However, they are primarily for internal use, which means that they are seldomly open-sourced and often poorly annotated. Consequently, unlike general-domain coding problems, natural science problems have less exposure in most current LMs' training data. This offers a unique opportunity to evaluate the models' ability to generalize to less familiar contexts. In total, SciCode consists of 80 main problems, decomposed into 338 subproblems.

Table 1 lists the subfields SciCode covers along with the number of main problems in each. Each main problem has a median of 3 subproblems, with a maximum of 15. We reserve 15 main problems (50 subproblems) for the development split and use the remaining 65 main problems (288 subproblems) as the test data. The 15 main development problems cover all five domains; over half of these have less than 4 subproblems each for easier few-shot settings.

| Fields | Subfields |
| --- | --- |
| Mathematics | Numerical linear Algebra (8), Computational Mechanics (5), Computational Finance (1) |
| Physics | Condensed Matter Physics (13), Optics (10), Quantum Information/Computing (6), Computational Physics (5), Astrophysics (2), Particle Physics (1) |
| Chemistry | Quantum Chemistry (5), Computational Chemistry (3) |
| Biology | Ecology (6), Biochemistry (1), Genetics (1) |
| Material Science | Semiconductor Materials (7), Molecular Modeling (6) |

Table 1: SciCode fields and subfields, with the number of main problems in each.

### 2.2 A Main Problem with Multiple Subproblems

In their everyday workflow, scientists often decompose a complex problem into multiple smaller, more manageable parts. They may write relatively independent code for each part and then integrate these parts into a complete solution to the main problem. In developing our dataset, we leverage

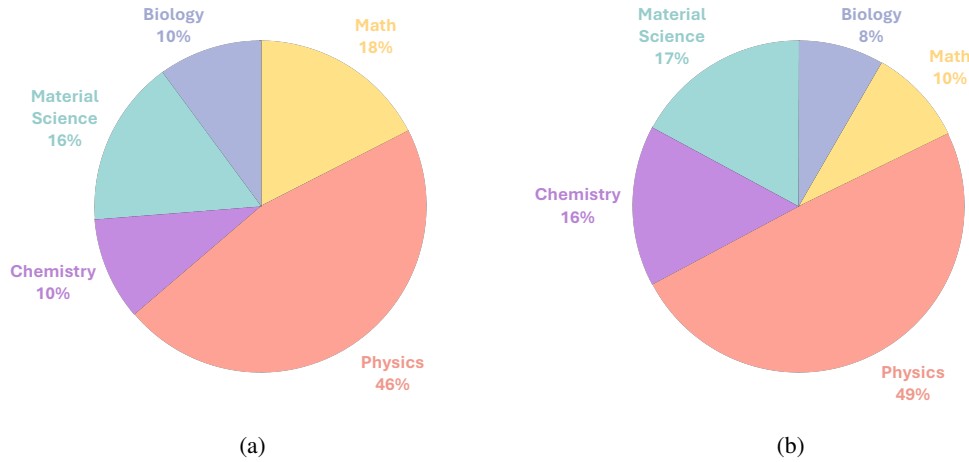

Figure 2: Distributions of (a) main problems and (b) subproblems.

this natural and intuitive structure and further standardize our dataset by instructing the scientists to adhere to the following format.

**Main Problem**   A *main problem* is a primary task that needs to be addressed. It defines the overall objective of the research and guides the direction of the study. The main problem encompasses all subproblems, with detailed instructions on required inputs and expected outputs articulated in a docstring block. With the main problem defined, scientists have sufficient guidance to solve the task.

**Subproblem Decomposition**   *Subproblems* focus on questions derived from the main problem. They decompose the complex main problem into smaller, more manageable parts, enabling a more detailed and systematic investigation. Detailed docstrings for each subproblem describe the required input and expected output, ensuring clarity and aiding in accurate code generation. This structured decomposition simplifies problem-solving and facilitates a more granular evaluation of the models' scientific coding capabilities.

### 2.3   Data Annotation

This process consists of three main stages:

(1) **Problem selection:** Deciding on question topics related to the research domain (§2.3.1).
(2) **Evaluation design:** Designing both numerical and domain-specific test cases to ensure the problem's validity (§2.3.2).
(3) **Problem validation:** Iterating on the problems through three rounds of revisions to further enhance question design (§2.3.3).

We now examine the design choices for each stage.

### 2.3.1   Problem Selection

Throughout the research project cycle, various coding needs arise, such as data processing, fitting, and plotting. To use SciCode, scientists select the problems that require intense scientific knowledge and reasoning to optimally test LM's science capability. This approach ensures that both the breadth and depth of frontier research are addressed. We focus on:

- **Numerical methods.** Analytical forms are usually impossible to achieve for very complicated systems. Therefore, scientists must derive numerical models and algorithms that describe physical phenomena [10, 76, 36, 79, 33], chemical reactions [23, 24, 89], biological systems [98, 99, 97, 85, 17, 16, 52, 91], or statistical behaviors[81, 71, 58, 56, 51, 25, 48].
- **Simulation of systems.** In fields of natural science, scientists write code to simulate systems and processes. These simulations are based on theoretical principles and empirical data,

reflecting deep scientific insights into the system being studied [78, 101, 63, 57, 75, 21, 92, 100, 39, 70, 42, 60, 73].

- **Scientific calculation.** During data post-processing and visualization, scientists often perform many transformations based on scientific formulas to get physical observable of interest instead of raw experimental data [11, 90, 31, 40, 41, 69, 6, 32].

We also include several research problems that are built upon or reproduce methods used in Nobel Prize-winning studies to highlight current trends in scientific research: the self-consistent field (SCF) method for density functional theory (DFT) calculations [38] **(The Nobel Prize in Chemistry 1998)**, the PMNS matrix for neutrino oscillation in matter [55, 62] **(The Nobel Prize in Physics 2015)**, the Haldane model for the anomalous quantum Hall effect [27] **(The Nobel Prize in Physics 2016)**, optical tweezer [47, 7] simulations for microscopic thermodynamics [51, 25, 48] **(The Nobel Prize in Physics 2018)**, and the replica method for spin glasses [81, 71, 58, 56] **(The Nobel Prize in Physics 2021)**.

### 2.3.2 Evaluation Design

To facilitate evaluation, we have scientist annotators use only widely adopted and well-documented packages such as NumPy, SciPy, and SymPy when writing the solution code for their problems, as shown in Figure 4.

Our test suite involves two key components. (1) **Numerical tests** list input-output pairs to check if the generated code produces the same outputs as ground truth. (2) **Domain-specific test cases**, introduced as an additional stage, evaluate whether model-generated solutions align with scientists' practical needs and further ensure the correctness and applicability of each solution within its specific field. These tests are extracted from real scientific workflows: scientists must design domain-specific test cases to verify code accuracy by reproducing results published in academic papers or matching analytical solutions derived from theoretical models. For example, we reproduce the phase transition at around $kT/J = 2.269$ for the 2D square Ising model problem [64], derive the surface plasmon mode in a 2D layered electron gas [11, 33], verify the ballistic Brownian motion in optical tweezer [47], etc. By doing so, we validate that the code not only functions correctly but also accurately represents the underlying scientific problem.

Overall, the evaluation design aims to balance the fidelity of the scientific problem with the practicality of the evaluation process, ensuring that the solutions are both accurate and accessible.

### 2.3.3 Problem Validation for Quality Control

We conduct three rounds of validation and revision for each problem:

(1) **In-domain scientist validation.** At least two scientists in the same research domain cross-check the *question design*, *solution code*, and *domain-specific test cases*, providing detailed feedback. The scientists who design the workflows iterate on them based on this feedback to ensure the problems are scientifically accurate.

(2) **Out-of-domain scientist validation.** One scientist from a different domain reviews the *question design* to ensure it is clear and that the information provided is precise and sufficient to solve the problem (e.g., all scientific constants are given). This helps to identify any assumptions that might be unclear to those outside the immediate field of study.

(3) **GPT-4 validation.** GPT-4 assists with the final review round. The previously validated sub-questions are input to GPT-4 to generate code solutions. Scientists perform error analysis for the generated solutions and redesign the *numerical test cases* if necessary to prevent false positives.Based on the code solutions from GPT-4, the scientist may also revise the *entire workflow* a third time to addressany potential ambiguity.

This multi-round validation approach ensures that the problems are scientifically rigorous, clear, and unambiguous, facilitating accurate and effective evaluation.

## 2.4 Various Types of Evaluations

SciCode offers unique opportunities for evaluating LMs across diverse settings, comprehensively testing their coding capabilities.

- **Without vs. with scientific background.** A subproblem can provide scientific background knowledge to guide LMs in solving the coding task. SciCode's scientific background for each problem offers two modes of evaluation. (1) When models are evaluated *without* scientific background, it tests their inherent scientific knowledge and reasoning along with their coding capability. (2) For models not designed to handle scientific problems, background provides the necessary knowledge and reasoning steps to solve the problems, shifting the evaluation's focus towards the models' coding and instruction-following capabilities. As we show in the experiments (§3), all models substantially improve performance when background is provided, indicating their lack of knowledge and reasoning capability in these natural science fields.
- **Gold vs. generated solutions to previous subproblems.** Each main problem in SciCode factorizes into multiple subproblems, and solutions to previous problems provide vital information for solving the current one. SciCode enables use of *gold* or *generated* solutions to previous subproblems. Gold solutions focus only on the current problem, while generated ones provide a more realistic evaluation setting and are more challenging due to error accumulation.
- **Main vs. subproblem levels.** (1) The LM is considered to have successfully solved the main problem when all subproblem solutions are correct and the integrated solution to the main problem is correct. (2) Alternatively, SciCode can assess at a subproblem level, evaluating a subproblem independently of other subproblems or its main problem.

Among these setups, evaluation *without background* carrying over *generated* solutions to previous problems is the closest to scientists' real use case of LMs. Therefore, we dub this the *standard setup*. Our experiments indicate that this setup is very challenging for even the best models available today: Claude3.5-Sonnet, the best performing one, can solve only 4.6% of the main problems.

To make SciCode useful for evaluating less capable or developing models, we also consider less challenging settings in our experiments.

## 3 Experiments

**Prompts.** We evaluate our model using zero-shot prompts. We keep the prompts general and design different ones for different evaluation setups only to inform the model about the tasks. We keep prompts the same across models and fields, and they contain the model's main and sub-problem instructions and code for previous subproblems. We also instruct the model to recall useful knowledge when gold background knowledge is not provided. §A.1 presents an example.

### 3.1 Evaluated Models

Since SciCode is a challenging benchmark, we mainly consider strong language models.[3]

- **OpenAI o1-preview** [67]: A new OpenAI model designed to spend more time thinking before they respond
- **OpenAI o1-mini** [67]: An efficient version of OpenAI o1-preview
- **GPT-4o** [66]: An optimized version of GPT-4 [65] by OpenAI with multi-modal capability.
- **GPT-4-Turbo**: A faster and more cost-effective variant of GPT-4 [65]. We use the 'gpt-4-turbo-2024-04-09' snapshot.
- **Claude3.5-Sonnet (new)** [3]: The upgraded model (20241022) from Claude3.5-Sonnet.
- **Claude3.5-Sonnet** [5]: The latest model from the Claude 3.5 family from Anthropic.
- **Claude3-Opus** [4]: The most capable model from the Claude 3 family from Anthropic.
- **Claude3-Sonnet** [4]: The second most capable model from the Claude 3 family.
- **Gemini 1.5 Pro** [87]: A model from the Gemini 1.5 family by Google and the largest with open access at the time of writing.
- **Llama-3-70B-Instruct** [2]: The instruction-tuned version of the largest available model from the Llama-3 family.
- **Llama-3.1-70B-Instruct** [2]: The instruction-tuned version of the largest available model from the Llama-3 family.

---

[3]For instance, CodeLlama-7B-Instruct achieves only 0.4% pass@1 in our main setting.

- **Llama-3.1-405B-Instruct** [2]: The instruction-tuned version of the largest available model from the Llama-3 family.
- **Mixtral-8x22B-Instruct** [34]: The instruction-tuned version of Mistral AI's largest publicly accessible Mixture-of-Expert Model.
- **Deepseek-Coder-v2** [104]: Mixture-of-Experts (MoE) code language model continue pre-trained on DeepSeek-V2
- **Qwen2-72B-Instruct** [88]: The largest instruction-tuned Qwen-2 model.

## 3.2 Main Results

| Models | Main Problem | Subproblem |
|---|---|---|
| *Proprietary Models* | | |
| OpenAI o1-preview | **7.7** | **28.5** |
| Claude3.5-Sonnet | 4.6 | 26.0 |
| Claude3.5-Sonnet (new) | 4.6 | 25.3 |
| GPT-4o | 1.5 | 25.0 |
| GPT-4-Turbo | 1.5 | 22.9 |
| OpenAI o1-mini | 1.5 | 22.2 |
| Gemini 1.5 Pro | 1.5 | 21.9 |
| Claude3-Opus | 1.5 | 21.5 |
| Claude3-Sonnet | 1.5 | 17.0 |
| *Open Models* | | |
| Deepseek-Coder-v2 | 3.1 | 21.2 |
| Llama-3.1-405B-Chat | 1.5 | 19.8 |
| Qwen2-72B-Instruct | 1.5 | 17.0 |
| Llama-3.1-70B-Chat | 0.0 | 17.0 |
| Mixtral-8x22B-Instruct | 0.0 | 16.3 |
| Llama-3-70B-Chat | 0.0 | 14.6 |

Table 2: Model performance in pass@1 rate on SciCode under the standard setup: without background knowledge and carrying over generated solutions to previous subproblems.

Table 2 presents results under the *standard setup*.[4] For the easier subproblem-level evaluation, the state-of-the-art models we test solve 14%-28.5% of the subproblems. Among them, OpenAI o1-preview achieves the best performance, with a 28.5% pass@1 rate. However, all models perform much worse on the more realistic and challenging main problem evaluation. OpenAI o1-preview still performs the best in this setting, but with only a 7.7% pass@1 rate.

These results show that SciCode is a difficult benchmark for current LMs. Consistent with our observations on proprietary models, open-weight LMs under test also showed their lack of capabilities in solving any main problem despite being able to solve a number of sub-problems correctly.

## 3.3 Additional Results with Other Evaluation Settings

**Providing gold scientific background knowledge.** Table 3 presents results when background text authored by scientists is provided to the LMs and generated solutions to previous subproblems are used. This setting evaluates both the models' capabilities to faithfully follow the instructions provided in the background as well as their code-generation performance. The $\Delta$ columns indicate performance differences compared to the standard setup.

---

[4]Without background and carrying over generated subproblem solutions. See §2.4 for a more detailed discussion.

*All models substantially improve performance for both subproblem and main problem evaluations when given scientific background knowledge.* For the subproblem evaluation, Claude3.5-Sonnet (new) performs the best with a 37.2% pass@1 rate. Llama-3.1-405B-Instruct benefits the most from the provided scientific background and reasoning with an increase of 13.2 %. However, Open models still improves less compared to proprietary models which might indicate weaker Instruction following capability. Interestingly, the comparison between Llama-3-70B-Instruct and Mixtral-8x22B-Instruct reveals a trend that differs from the standard setup: Llama-3-70B-Instruct benefits more from the scientific background knowledge and reaches the performance of Mixtral-8x22B-Instruct in this setting.

For the main problem evaluation, the trend remains similar to the standard setup. OpenAI o1-mini performs best, with a 13.8% pass@1 rate, followed closely by Claude3.5-Sonnet at 12.3%. OpenAI o1-mini improves most from background content, at 12.3%. Nonetheless, all models still fall short of satisfactory performance even with the background knowledge provided. This reaffirms that SciCode is challenging even when focusing on code generation rather than testing the models' scientific knowledge.

| Model | Main Problem | | Subproblem | |
|---|---|---|---|---|
| | Pass@1 | Δ | Pass@1 | Δ |
| *Proprietary Models* | | | | |
| OpenAI o1-mini | **13.8** | **12.3** | 34.4 | 12.2 |
| Claude3.5-Sonnet | 12.3 | 7.7 | 35.4 | 9.4 |
| Claude3.5-Sonnet (new) | 10.8 | 6.4 | **37.2** | 12.1 |
| OpenAI o1-preview | 10.8 | 3.1 | 34.0 | 5.5 |
| GPT-4o | 9.2 | 7.7 | 35.4 | 10.4 |
| GPT-4-Turbo-2024-04-09 | 9.2 | 7.7 | 33.7 | 10.8 |
| Gemini 1.5 Pro | 7.7 | 6.2 | 30.6 | 8.7 |
| Claude3-Opus | 4.7 | 3.0 | 26.7 | 5.2 |
| Claude3-Sonnet | 4.7 | 3.0 | 25.7 | 8.7 |
| *Open Models* | | | | |
| Llama-3.1-405B-Instruct | 10.8 | 9.3 | 33.0 | **13.2** |
| Deepseek-Coder-v2 | 4.6 | 1.5 | 27.1 | 5.9 |
| Llama-3.1-70B-Instruct | 4.6 | 4.6 | 25.7 | 8.7 |
| Qwen2-72B-Instruct | 4.6 | 3.1 | 22.2 | 5.2 |
| Mixtral-8x22B-Instruct | 3.1 | 3.1 | 20.8 | 4.5 |
| Llama-3-70B-Instruct | 1.5 | 1.5 | 20.8 | 6.3 |

Table 3: Pass@1 with generated solutions for previous subproblems and scientific background texts provided. The Δ columns show the performance differences compared to the *standard setting* in Table 2, i.e., where background content is *not* provided.

**With gold subproblem solutions.** Figure 3 plots the subproblem pass@1 rates conditioning on various numbers of previous subproblems and their gold solutions. Background knowledge is *not* provided. The intuition behind this analysis is that later steps can leverage gold solutions from previous steps to gain a richer understanding of the problem. Instructions and solutions from earlier steps serve as in-context demonstrations, enabling the model to rely less on its instruction-following capability. By focusing on later steps, we can more precisely assess the models' inherent capabilities.

*Overall, all three models show similar trends, with their performance generally improving as they condition on more gold solutions from previous steps.* However, there is a notable exception when conditioning on 7 previous gold subproblem solutions. Additionally, performance starts to decline when models condition on more than 9 previous solutions, possibly due to the increased difficulty of managing long contexts.

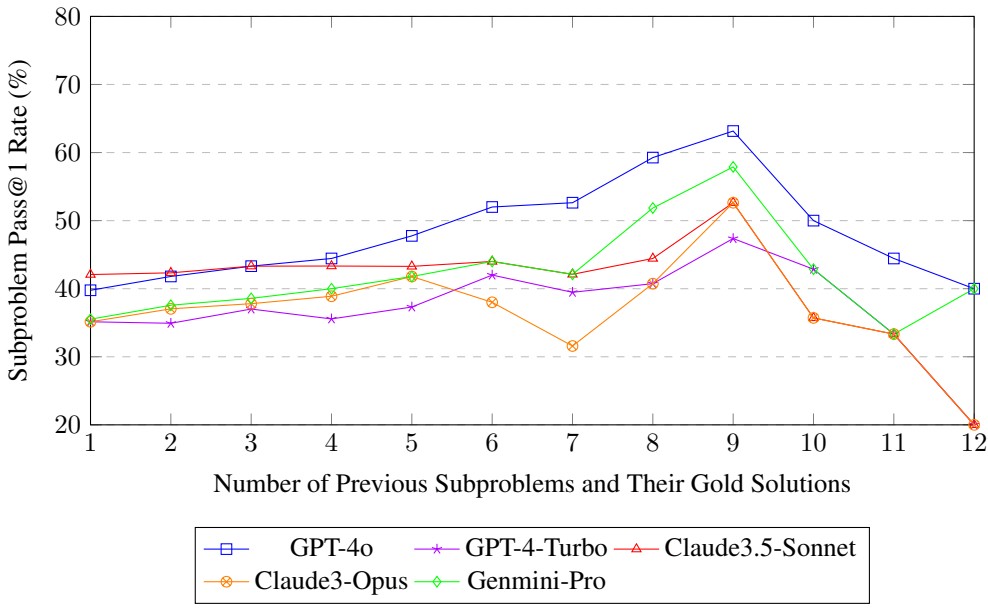

Figure 3: Subproblem pass@1 rate when conditioning on various numbers of previous subproblems and their gold solutions. Background knowledge is *not* provided.

## 4    Related Work

**Language models for code.**    Code has long been an active field of research, and code LMs have co-evolved with foundation LMs since the era of BERT [18]. Earlier works include CodeBert [20] and CodeT5 [96], while Codex [13] arguably kick-started the LLM era for code-generation models. Since Codex, the field has experienced rapid growth in quantity and quality of large code generation models, including specially trained models like Codegen [61], StarCoder models [46, 53], and generalist models with code adapation [13] such as CodeLlama [77], CodeQwen [9], and DeepSeek-Coder [26]. As code generation gains more attention and becomes increasingly useful, contemporary generalist models often include non-trivial coding capabilities [68, 87].

**Evaluating code generation.**    Before the emergence of very capable code synthesis models, when most models struggled to produce executable code, datasets like CoNaLa typically included n-gram-based metrics [102]. Soon after model capabilities improved, execution-based evaluation gained in popularity [29, 8, 12]. While n-gram or general text-based evaluation still exists, we opted to omit them from SciCode due to obvious limitations of surface form matching in scientific coding.

Code generation benchmarks now take various forms. For simple function completion, MBPP [8] and HumanEval [12] are two widely used benchmarks that contain basic programming questions, mainly evaluating LMs' ability to turn natural language instructions into Python programs. Other benchmarks assess the models' competence in real-world programming scenarios, such as writing data science code [43, 103], repository-level code completion [19], and more complex tasks in real-world software engineering [35]. Though our work is similar to MTPB [61] in terms of using a multi-turn setup, our subproblem instructions correspond to a high-level task, while theirs correspond to specific code actions (e.g., replace X with Y in the string).

**Language models for science.**    Scientific tasks are complex due to their demands for reasoning and knowledge. However, Recent advances in general and specialized language models have revolutionized the processing of text and other data modalities, such as molecules and proteins, in scientific fields. Galactica [86], a general-purpose scientific model, can perform tasks like citation prediction, scientific reasoning, document generation, and molecular property prediction. Many models focus on one single domain or task, like math (e.g., Minerva [44] and Deepseek-Math [80] ), protein structure prediction (e.g., ESM-2 [49]), medical reasoning (e.g., Med-PaLM [82], BioGPT [54]), and others.

## 5    Conclusion

We introduce SciCode, a scientific research benchmark curated by professional natural scientists. We designed SciCode for scientific problem evaluation and collected problems representing 16 diverse domains. By assessing SciCode with ten contemporary state-of-the-art AI models, we demonstrated that our benchmark is within reach but remains very challenging. We believe SciCode will serve as a helpful guideline for building future code language models for varied scientific applications.

## Acknowledgments and Disclosure of Funding

This work was supported by Laboratory Directed Research and Development (LDRD) funding from Argonne National Laboratory, provided by the Director, Office of Science, of the U.S. Department of Energy under Contract No. DE-AC02-06CH11357. The work used resources of the Argonne Leadership Computing Facility, a DOE Office of Science User Facility supported under Contract DE-AC02-06CH11357. EAH acknowledge support from National Science Foundation (NSF) award OAC-2209892. This research also used the Delta advanced computing and data resources, which is supported by the National Science Foundation (award OAC 2005572) and the State of Illinois. Delta is a joint effort of the University of Illinois Urbana-Champaign and its National Center for Supercomputing Applications. This research used the DeltaAI advanced computing and data resource, which is supported by the National Science Foundation (award OAC 2320345) and the State of Illinois. DeltaAI is a joint effort of the University of Illinois Urbana-Champaign and its National Center for Supercomputing Applications.

This work is in part supported by the Allen Institute for AI.

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

# A Appendix

## A.1 Prompt

| |
|---|
| **PROBLEM DESCRIPTION:**
You will be provided with problem steps along with background knowledge necessary for solving the problem. Your task will be to develop a Python solution focused on the next step of the problem-solving process.

**PROBLEM STEPS AND FUNCTION CODE:**
Here, you'll find the Python code for the initial steps of the problem-solving process. This code is integral to building the solution.
{problem_steps_str}

**NEXT STEP - PROBLEM STEP AND FUNCTION HEADER:**
This part will describe the next step in the problem-solving process. A function header will be provided, and your task is to develop the Python code for this next step based on the provided description and function header.
{next_step_str}

**DEPENDENCIES:**
Use only the following dependencies in your solution. Do not include these dependencies at the beginning of your code.
{dependencies}

**RESPONSE GUIDELINES:**
1. Start with the scientific background required for the next step, formatted as a comment.
2. Then write the complete and executable Python program for the next step in a single block.
3. Your response should focus exclusively on implementing the solution for the next step, adhering closely to the specified function header and the context provided by the initial steps.
4. DO NOT include previous function code, example usage or test code in your response.
5. Ensure your response is in the format of "'python"' and includes the necessary background as a comment at the top.
Example:
Background: [Here, insert the necessary scientific knowledge required for the next step.]
[Insert the Python code here based on the provided function header and dependencies.] |

Table 4: Prompt w/o Background

Table 5: Prompt w/ Scientists' Background

## A.2 Python libraries used in SciCode.

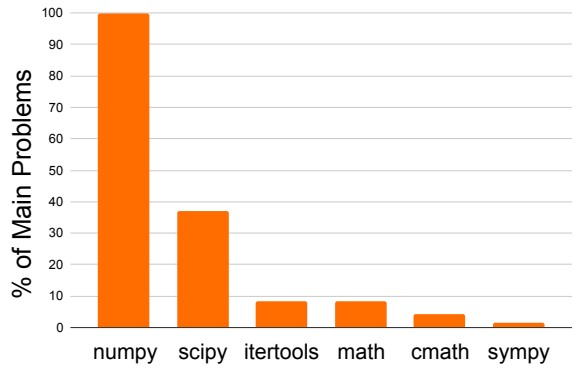

Figure 4: Python libraries used by problems in SciCode.

### A.3 SciCode Full Problem Example

#### A.3.1 Example Main Problem

> **Main Question**
>
> 1. Generate an array of Chern numbers for the Haldane model on a hexagonal lattice by sweeping the following parameters: the on-site energy to next-nearest-neighbor coupling constant ratio $(m/t_2)$ and the phase $(\phi)$ values. Given the lattice spacing $a$, the nearest-neighbor coupling constant $t_1$, the next-nearest-neighbor coupling constant $t_2$, the grid size $\delta$ for discretizing the Brillouin zone in the $k_x$ and $k_y$ directions (assuming the grid sizes are the same in both directions), and the number of sweeping grid points $N$ for $m/t_2$ and $\phi$.

> **Main Signature**
>
> ```
> Args:
> delta (float): The grid size in kx and ky axis for discretizing the Brillouin zone.
> a (float): The lattice spacing, i.e., the length of one side of the hexagon.
> t1 (float): The nearest-neighbor coupling constant.
> t2 (float): The next-nearest-neighbor coupling constant.
> N (int): The number of sweeping grid points for both the on-site energy to next-nearest-neighbor coupling constant ratio
> and phase.
>
> Returns:
> results (ndarray): 2D array of shape(N, N), the Chern numbers by sweeping the on-site energy to next-nearest-neighbor
> coupling constant ratio (m/t2) and phase (phi).
> m_values(ndarray): 1D array of length N, the swept on-site energy to next-nearest-neighbor coupling constant ratios.
> phi_values (ndarray): 1D array of length N, the swept phase values.
> ```

> **Dependencies**
>
> ```python
> import numpy as np
> import cmath
> from math import pi, sin, cos, sqrt
> ```

Figure 5: SciCode Example General Problem and Dependencies

#### A.3.2 Example Subproblems

> **Sub-Function 1 Question**
>
> 1.1 Write a Haldane model Hamiltonian on a hexagonal lattice, given the following parameters: wavevector components $k_x$ and $k_y$ (momentum) in the x and y directions, lattice spacing $a$, nearest-neighbor coupling constant $t_1$, next-nearest-neighbor coupling constant $t_2$, phase $\phi$ for the next-nearest-neighbor hopping, and the on-site energy $m$.

> **Sub-Function 1 Arguments**
>
> ```python
> def calc_hamiltonian(kx, ky, a, t1, t2, phi, m):
>     """
>     Function to generate the Haldane Hamiltonian with a given set of parameters.
>
>     Args:
>     kx (float): The x component of the wavevector.
>     ky (float): The y component of the wavevector.
>     a (float):  The lattice spacing, i.e., the length of one side of the hexagon.
>     t1 (float): The nearest-neighbor coupling constant.
>     t2 (float): The next-nearest-neighbor coupling constant.
>     phi (float): The phase ranging from -π to π.
>     m (float): The on-site energy.
>
>     Returns:
>     hamiltonian (ndarray): matrix of shape(2, 2) The Haldane Hamiltonian on a hexagonal lattice.
>     """
> ```

Figure 6: SciCode Example Subproblem 1

**Subproblem 1 Background**

Source: Haldane, F. D. M. (1988). Model for a quantum Hall effect without Landau levels: Condensed-matter realization of the" parity anomaly". Physical review letters, 61(18).

We denote $\{\mathbf{a}_i\}$ are the vectors from a B site to its three nearest-neighbor A sites, and $\{\mathbf{b}_i\}$ are next-nearest-neighbor distance vectors, then we have

$$\mathbf{a}_1 = (0, a),$$

$$\mathbf{a}_2 = \left(\frac{\sqrt{3}a}{2}, -\frac{a}{2}\right),$$

$$\mathbf{a}_3 = \left(-\frac{\sqrt{3}a}{2}, -\frac{a}{2}\right)$$

$$\mathbf{b}_1 = \mathbf{a}_2 - \mathbf{a}_3 = (\sqrt{3}a, 0),$$

$$\mathbf{b}_2 = \mathbf{a}_3 - \mathbf{a}_1 = \left(-\frac{\sqrt{3}a}{2}, -\frac{3a}{2}\right),$$

$$\mathbf{b}_3 = \mathbf{a}_1 - \mathbf{a}_2 = \left(-\frac{\sqrt{3}a}{2}, \frac{3a}{2}\right)$$

Then the Haldane model on a hexagonal lattice can be written as

$$H(k) = d_0 I + d_1 \sigma_1 + d_2 \sigma_2 + d_3 \sigma_3$$

$$d_0 = 2t_2 \cos\phi \sum_i \cos(\mathbf{k} \cdot \mathbf{b}_i)$$

$$= 2t_2 \cos\phi \left[\cos\left(\sqrt{3}k_x a\right) + \cos\left(-\frac{\sqrt{3}k_x a}{2} + \frac{3k_y a}{2}\right) + \cos\left(-\frac{\sqrt{3}k_x a}{2} - \frac{3k_y a}{2}\right)\right]$$

$$d_1 = t_1 \sum_i \cos(\mathbf{k} \cdot \mathbf{a}_i)$$

$$= t_1 \left[\cos\left(k_y a\right) + \cos\left(\frac{\sqrt{3}k_x a}{2} - \frac{k_y a}{2}\right) + \cos\left(-\frac{\sqrt{3}k_x a}{2} - \frac{k_y a}{2}\right)\right]$$

$$d_2 = t_1 \sum_i \sin(\mathbf{k} \cdot \mathbf{a}_i)$$

$$= t_1 \left[\sin\left(k_y a\right) + \sin\left(\frac{\sqrt{3}k_x a}{2} - \frac{k_y a}{2}\right) + \sin\left(-\frac{\sqrt{3}k_x a}{2} - \frac{k_y a}{2}\right)\right]$$

$$d_3 = m - 2t_2 \sin\phi \sum_i \sin(\mathbf{k} \cdot \mathbf{b}_i)$$

$$= m - 2t_2 \sin\phi \left[\sin\left(\sqrt{3}k_x a\right) + \sin\left(-\frac{\sqrt{3}k_x a}{2} + \frac{3k_y a}{2}\right) + \sin\left(-\frac{\sqrt{3}k_x a}{2} - \frac{3k_y a}{2}\right)\right]$$

where $\sigma_i$ are the Pauli matrices and $I$ is the identity matrix.

Table 6: Background Augmented by Scientists for Subproblem 1

## Sub-Function 2 Question

1.2 Calculate the Chern number using the Haldane Hamiltonian, given the grid size $\delta$ for discretizing the Brillouin zone in the $k_x$ and $k_y$ directions (assuming the grid sizes are the same in both directions), the lattice spacing $a$, the nearest-neighbor coupling constant $t_1$, the next-nearest-neighbor coupling constant $t_2$, the phase $\phi$ for the next-nearest-neighbor hopping, and the on-site energy $m$.

## Sub-Function 2 Arguments

```python
def compute_chern_number(delta, a, t1, t2, phi, m):
    """
    Function to compute the Chern number with a given set of parameters.

    Args:
    delta (float): The grid size in kx and ky axis for discretizing the Brillouin zone.
    a (float):  The lattice spacing, i.e., the length of one side of the hexagon.
    t1 (float): The nearest-neighbor coupling constant.
    t2 (float): The next-nearest-neighbor coupling constant.
    phi (float): The phase ranging from -π to π.
    m (float): The on-site energy.

    Returns:
    chern_number (float): The Chern number, a real number that should be close to an integer.
    The imaginary part is cropped out due to the negligible magnitude.
    """
```

Figure 7: SciCode Example Subproblem 2

---

**Subproblem 2 Background**

Source: Fukui, Takahiro, Yasuhiro Hatsugai, and Hiroshi Suzuki. "Chern numbers in discretized Brillouin zone: efficient method of computing (spin) Hall conductances." Journal of the Physical Society of Japan 74.6 (2005): 1674-1677.

Here we can discretize the two-dimensional Brillouin zone into grids with step $\delta k_x = \delta k_y = \delta$. If we define the U(1) gauge field on the links of the lattice as $U_\mu(\mathbf{k}_l) := \frac{\langle n(\mathbf{k}_l)|n(\mathbf{k}_l+\hat{\mu})\rangle}{|\langle n(\mathbf{k}_l)|n(\mathbf{k}_l+\hat{\mu})\rangle|}$, where $|n(\mathbf{k}_l)\rangle$ is the eigenvector of Hamiltonian at $\mathbf{k}_l$, $\hat{\mu}$ is a small displacement vector in the direction $\mu$ with magnitude $\delta$, and $\mathbf{k}_l$ is one of the momentum space lattice points $l$. The corresponding curvature (flux) becomes

$$F_{xy}(\mathbf{k}_l) := \ln\left[U_x(\mathbf{k}_l)U_y(\mathbf{k}_l+\hat{x})U_x^{-1}(\mathbf{k}_l+\hat{y})U_y^{-1}(\mathbf{k}_l)\right]$$

and the Chern number of a band can be calculated as

$$c = \frac{1}{2\pi i}\sum_l F_{xy}(\mathbf{k}_l),$$

where the summation is over all the lattice points $l$. Note that the Brillouin zone of a hexagonal lattice with spacing $a$ can be chosen as a rectangle with $0 \le k_x \le k_{x0} = \frac{2\sqrt{3}\pi}{3a}, 0 \le k_y \le k_{y0} = \frac{4\pi}{3a}$.

Table 7: Background Augmented by Scientists for Subproblem 2

## Sub-Function 3 Arguments

```
def compute_chern_number_grid(delta, a, t1, t2, N):
    """
    Function to calculate the Chern numbers by sweeping the given set of parameters and returns the results along
    with the corresponding swept next-nearest-neighbor coupling constant and phase.

    Args:
    delta (float): The grid size in kx and ky axis for discretizing the Brillouin zone.
    a (float):  The lattice spacing, i.e., the length of one side of the hexagon.
    t1 (float): The nearest-neighbor coupling constant.
    t2 (float): The next-nearest-neighbor coupling constant.
    N (int):    The number of sweeping grid points for both the on-site energy to next-nearest-neighbor coupling
    constant ratio and phase.

    Returns:
    results (ndarray): 2D array of shape(N, N), The Chern numbers by sweeping the on-site energy to next-nearest-
    neighbor coupling constant ratio (m/t2) and phase (phi).
    m_values (ndarray): 1D array of length N, The swept on-site energy to next-nearest-neighbor coupling constant
    ratios.
    phi_values (ndarray): 1D array of length N, The swept phase values.
    """
```

Figure 8: SciCode Example Subproblem 3

### A.3.3   Example Domain Specific Test Cases

Both the $k$-space and sweeping grid sizes are set to very rough values to make the computation faster, feel free to increase them for higher accuracy.

At zero on-site energy, the Chern number is 1 for $\phi > 0$, and the Chern number is -1 for $\phi < 0$.

For complementary plots Figure 9, we can see that these phase diagrams are similar to the one in the original paper: Fig.2 in Haldane, F. D. M. (1988). To achieve a better match, decrease all grid sizes.

Compare the following three test cases. We can find that the phase diagram is independent of the value of $t_1$, and the ratio of $t_2/t_1$, which is consistent with our expectations.

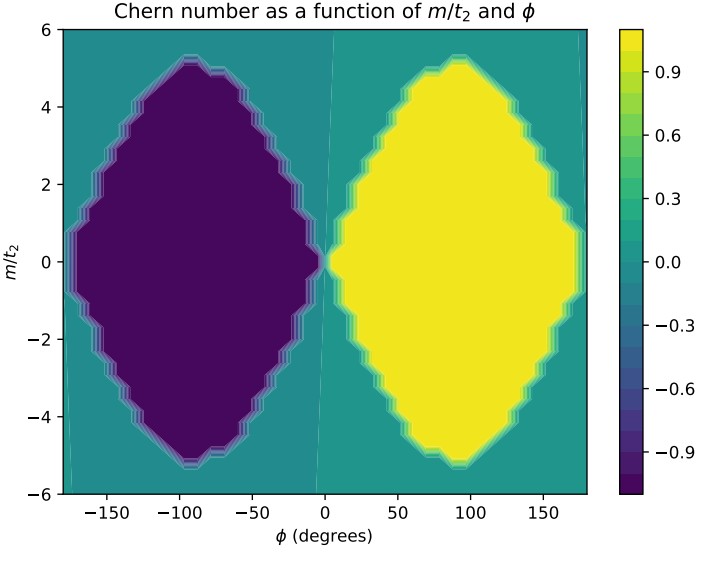

(a) delta = $2\pi/30$, a = 1.0, t1 = 4.0, t2 = 1.0, N = 40

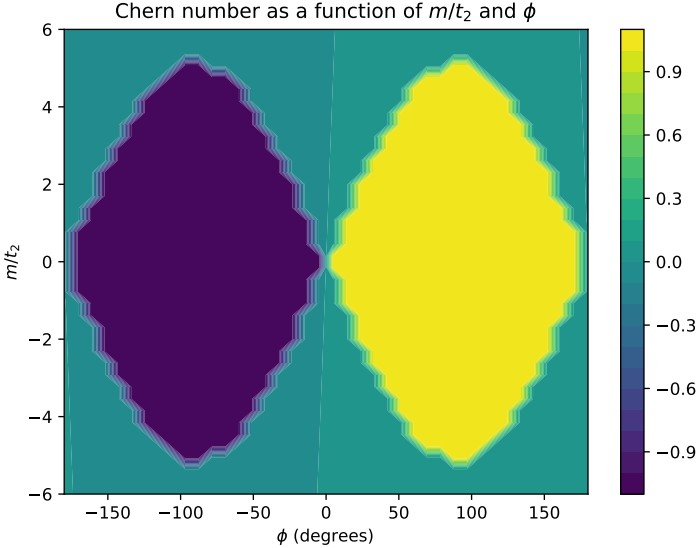

(b) delta = $2\pi/30$, a = 1.0, t1 = 5.0, t2 = 1.0, N = 40

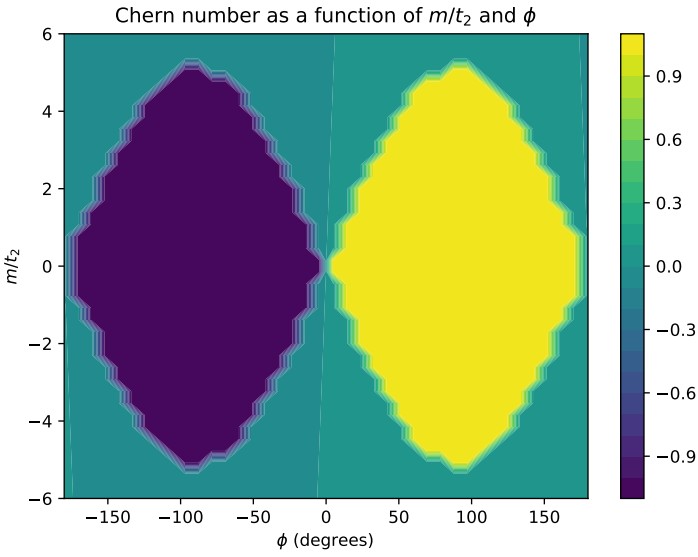

(c) delta = $2\pi/30$, a = 1.0, t1 = 1.0, t2 = 0.2, N = 40

Figure 9: Complementary Figures of Domain Specific Test Cases

