# OpenReview forum: "SciCode: A Research Coding Benchmark Curated by Scientists"
_NeurIPS.cc/2024/Datasets_and_Benchmarks_Track — NeurIPS 2024 Track Datasets and Benchmarks Poster_

### Official Review · Reviewer_SHks · 2024-07-06
**A valuable high-quality dataset for expert-level scientific programming tasks**

**Rating:** 8
**Confidence:** 4
**Clarity:** No complaints. Paper is well-written …

**Review:**

Quality: Looks well-made and carefully thought through. The dataset appears to have been sourced carefully with thorough quality control. The dataset contains relevant and useful tasks representing real-world challenges that scientists have to solve.

Clarity: The paper is well-written and easy to understand.

Significance: It is exciting to see a new expert-level benchmark where contemporary models presently fail (best models achieve <5% on the standard setup). High-quality expert-level datasets are crucial for pushing progress in the field forward, and I believe this will be a valuable indicator of progress in language models for science.

**Strengths:**

Relevant and useful dataset content
- SciCode is realistic and sources problems from scientists’ everyday research tasks or important papers.
- Including problems from Nobel Prize-winning studies is an exciting addition.

Thoughtful problem setup
- Having each Main Problem be decomposable into smaller Subproblems allows for high-resolution progress tracking (compared to the alternative of having just main-problem questions). Decomposition also clearly demonstrates how complex each Main Problem is.
- SciCode provides gold-standard solutions and multiple test cases for each problem. Test cases include a variety of automatic numerical tests and domain-specific tests, such that submitted answers can be evaluated in a completely automated fashion.

Annotation quality
- High annotation quality: all problems are annotated, revised, and verified by at least two researchers at the senior PhD student level or above.
- No data contamination: All SciCode problems are revised by annotators to ensure zero overlap with publicly available datasets.

Relevance
- Very valuable expert signal in this dataset. It is difficult and expensive to collect expert-level datasets, yet crucial for measuring progress. This will be of great interest to groups building and evaluating language models in scientific applications.
- Room for progress in the field: SciCode is a very challenging benchmark. The strongest model in the experiments only solve 3.4% of the main problems, highlighting an opportunity for the field to make progress in this direction.

**Additional Feedback:**

Typo on Page 5 (Evaluation Design): "To facilitate evaluation, we restrict annotators to using only utilizing only 149 widely-used and well-documented packages"

Thank you for your work!

**Correctness:**

The dataset construction and benchmark setup is thoughtful and well-executed.

In evaluation results, it would be nice to have shown few-shot results instead of just zero-shot, especially given that the authors already mentioned having suitable examples for few-shot in the development set. The relevance of this benchmark depends to some extent on how challenging it is for existing models, so it will be interesting to try to push the limits of current language models against this. That said, I expect the community to do this after the benchmark is released so it is fine for the authors to have focused primarily on the dataset.

It would also be useful to have error bars via multiple repeats, especially given the small dataset size. When it is shown that GPT-4o scores 3.4%, that's actually just 2/58 on the test set. Presenting differences between models as 0% vs 1.7% vs 3.4% can be misleading when the difference of 1.7% comes from just a single sample of difference on likely high-variance inference procedures.

**Documentation:**

The data collection process is well-documented in section 2.3. The authors provide a link to the dataset, with the data files provided as-is without further documentation; however the dataset is straightforward and cleanly structured so it is clear how to use the dataset based on the setup in the paper. There is sufficient information for reproducibility.

**Ethics:**

No ethical concerns.

**Limitations:**

No clear discussion of limitations in the paper, but no clear limitations I can think of either.

**Opportunities For Improvement:**

1. Could do with better analysis of failures - why do models do so poorly here? Is it because they lack domain knowledge, fail to write code correctly, have failures of executive function, or something else? In "Problem Validation", the authors mention "perform error analysis for the generated solutions". Some of the analyses might be interesting to share.

2. In Figure 3, the authors note that "performance starts to decline when models condition on more than 10 previous solutions, possibly due to the increased difficulty of managing long contexts." This seems like a surprising and abrupt drop-off in performance for all models (which have different context lengths). Could we get more certainty about what's going on here?

3. Big fan of the "Background knowledge" experiment! However, it's unclear whether the background knowledge is sufficient to plug the gaps in scientific knowledge. For example, if an expert human coder / scientist (from a different domain) is given the background knowledge, should we expect them to score 100% on the problems?

**Relation To Prior Work:**

I view this benchmark to be at the intersection of benchmarks for language models in 1) coding and 2) scientific knowledge / application. There is reasonable discussion of prior work on benchmarking language models for coding. However, the Related Work section does not compare their work against other benchmarks on scientific knowledge.

**Summary And Contributions:**

The SciCode benchmark provides a dataset of 73 challenging and realistic scientific programming problems spanning math, physics, chemistry, biology, and material science. The problems are written by human scientists, requiring expert-level scientific knowledge and programming ability to solve. Each of the 73 main problems can be decomposed into up to 15 subproblems, each with gold solutions and requisite background knowledge to allow for detailed analysis and a variety of evaluation setups.

---

> ### Author Rebuttal · Authors · 2024-08-20
>
> ***1. Could do with better analysis of failures - why do models do so poorly here? Is it because they lack domain knowledge, fail to write code correctly, have failures of executive function, or something else? In "Problem Validation", the authors mention "perform error analysis for the generated solutions". Some of the analyses might be interesting to share.***
>
> We thank the reviewer for the great suggestion. We will add a more detailed error analysis section in the revision. We have performed error analysis at subproblems with 3 types of reasons: ‘Lack of Knowledge/Misunderstand Question’, ‘Not following instruction’, and ‘Wrong Calculation/Methodology’. ‘Wrong Calculation/Methodology’ is the most common followed by ‘Lack of Knowledge/Misunderstand Question’ and ‘Not following instruction’.
>
> ***2. In Figure 3, the authors note that "performance starts to decline when models condition on more than 10 previous solutions, possibly due to the increased difficulty of managing long contexts." This seems like a surprising and abrupt drop-off in performance for all models (which have different context lengths). Could we get more certainty about what's going on here?***
>
> The average context length at the declining (11th) step is approximately 5,600 tokens. Upon closer examination of the domain distribution for longer problems (those exceeding 10 steps), we find that Physics dominates this category, with 11 problems, compared to 1 from Chemistry and 2 from Material Science. Among the domains that tend to have long, multistep problems (i.e., Material Science, Chemistry, and Physics), Physics poses the greatest challenge for LLMs to solve with the lowest accuracy. As a result, this skewed domain distribution likely contributes to the observed performance decline. We will add this discussion in the revision.
>
> ***3. Big fan of the "Background knowledge" experiment! However, it's unclear whether the background knowledge is sufficient to plug the gaps in scientific knowledge. For example, if an expert human coder / scientist (from a different domain) is given the background knowledge, should we expect them to score 100% on the problems?***
>
> We thank the reviewer for raising this question, and are glad that they like our design choice on the background knowledge! The question along with background knowledge can be sufficient for out-of-domain scientists to solve the problem: in the second round of validation for each problem, out-of-domain scientists are asked to verify that the problem can be solved by scientists in a relevant but different field, given the background knowledge. We do not have enough evidence to conclusively claim that all expert humans are able to solve the problem with 100% success rate since we have not verified this with large-scale human evaluation, which can be extremely expensive and beyond our scope.

---

> ### Author Rebuttal · Authors · 2024-08-31
>
> Dear Reviewers,
>
> Thank you for your constructive feedback. We have provided additional experimental results in our response. Since it is close to the end of the discussion period, please let us know if we should include anything further in the revised draft. We are more than happy to clarify if anything is unclear.
>
> Best regards,
>
> Authors

---

### Official Review · Reviewer_fuak · 2024-07-23
**SciCode: A Research Coding Benchmark Curated by Scientists**

**Rating:** 5
**Confidence:** 4
**Correctness:** Yes
**Clarity:** Yes

**Review:**

The paper explores a relatively complex and important problem of scientific coding. This seems to be relatively under-researched due to the manual effort and complexity required to collect this data. Scientific coding benchmarks like these also seem to be an important step towards evaluating the reasoning abilities of LLMs.

Pros:
1. Developing a scientific coding benchmark is a significant challenge and a step towards understanding an important aspect of how LLMs reason.
2. The benchmark contains a diverse set of problems and subproblems from multiple subfields of natural science which require significant effort to collect and annotate.
3. The benchmark also evaluates several different LLMs in its evaluation, showing how effective they are at solving individual subproblems versus the entire problem itself.

Cons:
1. The evaluation lacks a diversity of prompting techniques and models that are shown to reason better than vanilla LLMs, like neurosymbolic methods, chain-of-thought (or even faithful chain-of-thought) reasoning, and others. I don't believe that they would do much better for such complex problems, but I would regardless like to see their performance in such cases.
2. There is a lack of details in the construction of the dataset itself, like describing who chose the problems and divided them into subproblems, how many scientists and researchers were involved in total, was there any overlap between the scientists who performed quality control and those who designed the problems, and a discussion about problems that could be solved in multiple ways.
3. A portion of the validation is performed with GPT-4. I am concerned that this may bias the dataset towards favoring the other models from the same family (like GPT-4-o and GPT-4 Turbo).

Additional Questions and Suggestions
1. Typo at line 148 : " to using only utilizing only..."
2. Can you give more information about the problems that were disqualified after quality testing, as well as the number of those problems?
3. Typo at line 201 : "suceessfully"
4. What is the length of contexts in the case of the examples in Figure 3 where the accuracy starts declining? Also how many subproblems can a context consider at a time?

**Strengths:**

See pros.

**Additional Feedback:**

NA

**Documentation:**

The README of the provided data is empty.

**Limitations:**

It would be nice to have a dedicated limitations section that describes the general limitations of the system and also addresses the concerns raised in the review.

**Opportunities For Improvement:**

See cons and additional questions.

**Relation To Prior Work:**

Yes

**Summary And Contributions:**

This paper proposes SciCode, a benchmark consisting of scientific coding facts from multiple fields, with 73 problems decomposed into 305 subproblems. These are complex tasks which are unfamiliar to LLMs, and annotated by scientists and researchers. The paper demonstrates an evaluation of 8 models over SciCode with several different setups to show the levels of reasoning abilities of these models.

---

> ### Author Rebuttal · Authors · 2024-08-20
>
> ***1. The evaluation lacks a diversity of prompting techniques and models that are shown to reason better than vanilla LLMs, like neurosymbolic methods, chain-of-thought (or even faithful chain-of-thought) reasoning, and others. I don't believe that they would do much better for such complex problems, but I would regardless like to see their performance in such cases.***
>
> We thank the reviewer for the great suggestion. We prioritize using simple prompt techniques to focus on evaluating the instruction-following capabilities of advanced models like GPT-4o, Claude 3, Gemini 1.5 Pro, and Llama 3, as their technical reports emphasize strong reasoning abilities, which is our key area of interest. Due to the limited turn around time for the responsetime constraints, we are able to conducted a zero-shot chain-of-thought evaluation on GPT-4o, GPT-4 Turbo, and Claude 3.5 Sonnet. We found minimal impact on performance. More specifically, we observeThere was only a 1% improvement in the standard subproblem setup and no noticeable improvement in the main problem setup, suggesting that CoT techniques do not always enhance model performance.
> In the revision, we plan to finish this zero-shot CoT experiment with other models, and explore few-shot CoT.
>
>
> ***2.There is a lack of details in the construction of the dataset itself, like describing who chose the problems and divided them into subproblems, how many scientists and researchers were involved in total, was there any overlap between the scientists who performed quality control and those who designed the problems, and a discussion about problems that could be solved in multiple ways.***
>
> The annotators of SciCode are 20 scientists at the senior PhD level or above in their respective fields of study. To ensure that SciCode realistically reflects the application scenarios of language models, we rely on these expert annotators to determine the topics crucial to their areas of expertise and to select the problems based on their own research experience. They are also responsible for breaking down the main problems into subproblems.
>
> For quality control, there is no overlap between the scientists who design a problem and those who validate it. Each problem undergoes two rounds of validation. In the first round, the problem is proofread by at least two scientists from the same subdomain, and then revised based on their feedback. In the second round, the problem is proofread by scientists from the same domain but outside the immediate subdomain. For example, a problem in optics is first checked by two scientists in that field and, after revision, is reviewed by scientists from related areas such as astrophysics and computational physics.
>
> The reviewer rightly pointed out that some problems can be solved in multiple ways. We believe this has little impact on our evaluation process because, rather than comparing the models’ outputs to gold-standard solutions, SciCode evaluates using test cases by executing the solutions and checking the correctness of the results. Therefore, a solution is considered correct as long as it produces the correct results on the test cases, even if it differs from our gold-standard solutions.
> 	We will clarify these points in the revision.
>
> ***3. A portion of the validation is performed with GPT-4. I am concerned that this may bias the dataset towards favoring the other models from the same family (like GPT-4-o and GPT-4 Turbo).***
>
> We appreciate the reviewer for raising this concern. The final GPT-4 validation is implemented to address ambiguity and ensure quality control, rather than to simplify questions for LLMs. We have carefully instructed scientists and monitored the process through direct discussions with them. Therefore, we believe that SciCode does not favor the GPT models over others due to their minimal  involvement in the data collection process:
> Detecting false positive solutions due to numerical issues: Most problems in SciCode require floating-point number tensors as the execution results. Therefore, accounting for numerical precision issues is crucial. We use numpy.allclose with default tolerance to determine whether two tensors—one generated by executing the code and the other being the gold standard—are equivalent. This method can, inevitably, result in some false positive solutions, where the generated results are incorrect but still pass the tests due to the small magnitude of differences. GPT-4 assists our annotators in identifying these cases, allowing them to design stricter test cases or, in some instances, revise the problems.
>  Adding scientific constants/units to prevent ambiguity: GPT-4 is used to identify cases where scientific constants or units are not specified, which can lead to ambiguity. For example, Planck constant is usually expressed in 2 forms: 6.626E-34 J*s or 4.136 eV*s which are both correct.  In such cases, multiple solutions might be correct but differ in the units used. This ambiguity makes evaluation more challenging. Therefore, GPT-4 helps annotators identify and eliminate these ambiguities.
> In sum, **SciCode never uses GPT-4 models to filter or reword any problems**; all revisions are done by our scientists. We believe that SciCode does not favor GPT-4 or any language model, as evidenced by our results showing that GPT-4 underperforms Claude-Opus.

---

> > ### Author Rebuttal · Authors · 2024-08-20
> >
> > ***Additional Questions and Suggestions***
> >
> > ***1. Typo at line 148 : " to using only utilizing only..."***
> >
> > Thank you for pointing out! We will correct this in the version.
> >
> >
> > ***2. Can you give more information about the problems that were disqualified after quality testing, as well as the number of those problems?***
> >
> > We have 10 problems (~60 subproblems) that we decided not to include in the final set due to various reasons. Some of the problem results are not reproducible due to the inclusion of the stochastic process. Some of the problems take too long (more than half an hour for one test case) to simulate physically reliable results. Sometimes, scientists cannot find a good/easy way to evaluate the correctness of the results due to complex setup, we also do not include those problems.
> >
> > ***3. Typo at line 201 : "suceessfully"***
> >
> > Thank you for pointing out! We will correct this in the version.
> >
> > ***4. What is the length of contexts in the case of the examples in Figure 3 where the accuracy starts declining? Also how many subproblems can a context consider at a time?***
> >
> > The average context length at the declining (11th) step is around 5600. This includes 10 previous questions and code pairs.

---

> ### Author Rebuttal · Authors · 2024-08-31
>
> Dear Reviewers,
>
> Thank you for your constructive feedback. We have provided additional experimental results in our response. Since it is close to the end of the discussion period, please let us know if we should include anything further in the revised draft. We are more than happy to clarify if anything is unclear.
>
> Best regards,
>
> Authors

---

### Official Review · Reviewer_qWyP · 2024-07-29
**Good paper, some additional case study experiments would help**

**Rating:** 8
**Confidence:** 4
**Correctness:** See Section "Review"
**Clarity:** See Section "Review"

**Review:**

> [93-95] We reserve 15 main problems (58 subproblems) for the development split , and use the remaining 58 main problems (247 subproblems) as the test data.

I would suggest having a test/val/few shot separation with a hidden benchmark to avoid data contamination for future models.

> [185-186] When necessary, a subproblem also provides optional scientific background knowledge to guide the models in solving the coding task. This optional

I would have really liked to see a distribution over the requisite background knowledge & sub-task. Is it string manipulation? Is it loop construction? Is it constructing conditionals?

> [213] We evaluate our model using zero-shot prompt. We keep the prompts general, and only

I do not expect authors to use complicated prompts but I would like to see a discussion of using CoT/self-consistency with an example of whether model performance improves/decreases [I do not think it will always improve, which is why would be an interesting "case study" example.]

> [232-233] Table 2: Model performance in pass@1 rate on SciCode under the standard setup: without background knowledge and carrying over generated solutions to previous subproblems.

You can improve this table considerably showing the distribution of model perf over different categories of questions and knowledge requisite classification.

> [233] Table 2 presents the results under the standard setup.

For the easier subproblem level evaluation, the state-of-the-art models we test manage to solve 20%-30% of the subproblems. Among them, Claude3-Opus and Gemini 1.5 Pro achieve the best performance with a 29.6% pass@1 rate. However, all models achieve much worse performance on the more realistic and challenging main problem level evaluation. GPT-4o performs the best in this setting, with only a 3.4% pass@1 rate, and Gemini 1.5 Pro 1.7%. All other models cannot solve any main problem, including Claude3-Opus, despite its decent performance in the subproblem level evaluation. These results indicate that SciCode is a very challenging benchmark for current LMs. Consistent with our observations on proprietary models, open-weight language models under test also showed their lack of capabilities in solving any main problem, while being able to solve a number of sub-problems correctly.

> [257-258] Interestingly, although Claude3-Opus benefits the most from the background in the subproblem level evaluation, it gains less in the main problem evaluation.

This is why I think it would be helpful to do tables that are {model} x {background} x {task level} x {field/science} x {background knowledge type} x {coding task type}

> [263] With Gold Subproblem Solutions,

Interesting conditional that I would not have thought to include but makes perfect sense to have.

**Strengths:**

See Section "Review"

**Additional Feedback:**

N/A

**Documentation:**

It says that the code and data are proprietary, but other than that the prompts have been provided for replication of results, which I think is fair.

**Limitations:**

See Section "Review"

**Opportunities For Improvement:**

See Section "Review"

**Relation To Prior Work:**

See Section "Review"

**Summary And Contributions:**

Most coding datasets are small and focused on particular reactions derived from basic tasks like string manipulation. Current models don't perform well on these datasets, even with recent studies looking at dataset (HumanEval) leakages based on semantic similarities or inclusion in training data. This paper proposes using difficult problems from various scientific coding domains to reach solutions, performing different tasks that utilize either the final answer or responses from sub-problems. I believe this dataset would be valuable for assessing how far we've progressed in developing models capable of independently coding useful structures.

---

> ### Author Rebuttal · Authors · 2024-08-20
>
> > [93-95] We reserve 15 main problems (58 subproblems) for the development split , and use the remaining 58 main problems (247 subproblems) as the test data.
>
> ***I would suggest having a test/val/few shot separation with a hidden benchmark to avoid data contamination for future models.***
>
> Data contamination is definitely something in our consideration. Therefore, when releasing the dataset, we only released the question and annotated background as well as the final outputs but not the gold solution. We appreciate the reviewer’s great suggestion on the hidden benchmark.  We are continually collecting more evaluation problems for SciCode, and will consider building a hidden benchmark using the newly-collected problems.
>
> > [185-186] When necessary, a subproblem also provides optional scientific background knowledge to guide the models in solving the coding task. This optional
>
> ***I would have really liked to see a distribution over the requisite background knowledge & sub-task. Is it string manipulation? Is it loop construction? Is it constructing conditionals?***
>
> The scientific background for these subproblems does not fit neatly into typical coding categories like string manipulation, loop construction, or conditional statements. Instead, we instruct scientists to provide a problem background designed for individuals with general scientific training but lacking domain-specific expertise, ensuring they receive sufficient guidance to approach the task. This background information combines scientific knowledge, formula derivation, and pseudo-algorithm construction, making it a natural blend tailored for the problem at hand. The process has been monitored, revised, and validated by domain scientists and further tested by out-of-domain scientists to ensure it mimics real-world conditions effectively.
>
> For example, in our SciCode example problem in figure 1, for question:
>
> **1.2 Calculate the Chern number using the Haldane Hamiltonian, given the grid size $\delta$ for discretizing the Brillouin zone in the $k_x$ and $k_y$ directions (assuming the grid sizes are the same in both directions), the lattice spacing $a$, the nearest-neighbor coupling constant $t_1$, the next-nearest-neighbor coupling constant $t_2$, the phase $\phi$ for the next-nearest-neighbor hopping, and the on-site energy $m$.**
>
> We have the scientist annotated background:
>
> Here we can discretize the two-dimensional Brillouin zone into grids with step $\delta {k_x} = \delta {k_y} = \delta$. If we define the U(1) gauge field on the links of the lattice as $U_\mu (\mathbf{k}_l) := \frac{\left\langle n(\mathbf{k}_l)\middle|n(\mathbf{k}_l + \hat{\mu})\right\rangle}{\left|\left\langle n(\mathbf{k}_l)\middle|n(\mathbf{k}_l + \hat{\mu})\right\rangle\right|}$, where $\left|n(\mathbf{k}_l)\right\rangle$ is the eigenvector of Hamiltonian at $\mathbf{k}_l$, $\hat{\mu}$ is a small displacement vector in the direction $\mu$ with magnitude $\delta$, and $\mathbf{k}_l$ is one of the momentum space lattice points $l$. The corresponding curvature (flux) becomes
>
>
> $$
> F_{xy}(\mathbf{k}_l) := \ln \left[U_x(\mathbf{k}_l)U_y(\mathbf{k}_l+\hat{x})U_x^{-1}(\mathbf{k}_l+\hat{y})U_y^{-1}(\mathbf{k}_l)\right]
> $$
>
>
> and the Chern number of a band can be calculated as
>
>
> $$
> c = \frac{1}{2\pi i} \Sigma_l F_{xy}(\mathbf{k}_l),
> $$
>
> where the summation is over all the lattice points $l$. Note that the Brillouin zone of a hexagonal lattice with spacing $a$ can be chosen as a rectangle with $0 \le {k_x} \le k_{x0} = 2\sqrt 3 \pi /(3a),0 \le {k_y} \le k_{y0} = 4\pi /(3a)$.
>
> The background in this problem not only introduces the discretization of the Brillouin zone into grids but also provides a strategy for calculating the Chern number using this approach.
>
> > [213] We evaluate our model using zero-shot prompt. We keep the prompts general, and only
>
> ***I do not expect authors to use complicated prompts but I would like to see a discussion of using CoT/self-consistency with an example of whether model performance improves/decreases [I do not think it will always improve, which is why would be an interesting "case study" example.]***
>
> Thank you for the great suggestion. We prioritize using simple prompt techniques to focus on evaluating the instruction-following capabilities of advanced models like GPT-4o, Claude 3, Gemini 1.5 Pro, and Llama 3, as their technical reports emphasize strong reasoning abilities, which is our key area of interest. Due to the limited turn around time for the response, we are able to conduct a zero-shot chain-of-thought evaluation on GPT-4o, GPT-4 Turbo, and Claude 3.5 Sonnet. We found minimal impact on performance. More specifically, we observe only a 1% improvement in the standard subproblem setup and no noticeable improvement in the main problem setup, suggesting that CoT techniques do not always enhance model performance.
>
> In the revision, we plan to finish this zero-shot CoT experiment with other models, and explore few-shot CoT.

---

> > ### Author Rebuttal · Authors · 2024-08-20
> >
> > > [232-233] Table 2: Model performance in pass@1 rate on SciCode under the standard setup: without background knowledge and carrying over generated solutions to previous subproblems.
> >
> > ***You can improve this table considerably showing the distribution of model perf over different categories of questions and knowledge requisite classification.***
> >
> > Thank you for the suggestion! Here we include an extension of table 2 with domain categories.
> > | Model                          | Math (%) | Physics (%) | Chemistry (%) | Biology (%) | Material Science (%) |
> > |---------------------------------|----------|-------------|---------------|-------------|----------------------|
> > | GPT-4-turbo-2024-04-09          | 12.50    | 25.33       | 13.51         | 16.00       | 29.09                |
> > | GPT-4o                          | 8.33     | 24.67       | 27.03         | 24.00       | 30.91                |
> > | Claude-3-sonnet-20240229        | 8.33     | 18.67       | 16.22         | 12.00       | 18.18                |
> > | Claude-3-opus-20240229          | 8.33     | 21.33       | 18.92         | 24.00       | 27.27                |
> > | Claude-3-5-sonnet-20240620      | 12.50    | 26.00       | 27.03         | 32.00       | 27.27                |
> > | Gemini-1.5-pro                  | 8.33     | 21.33       | 18.92         | 24.00       | 29.09                |
> > | Llama-3-70b-chat-hf             | 8.33     | 16.00       | 2.70          | 16.00       | 20.00                |
> > | Mixtral-8x22B-Instruct-v0.1     | 8.33     | 17.33       | 8.11          | 12.00       | 23.64                |
> > | Qwen2-72B-Instruct              | 8.33     | 20.00       | 8.11          | 20.00       | 16.36                |
> > | Deepseek-coder                  | 8.33     | 21.33       | 2.70          | 28.00       | 34.55                |
> >
> > > [257-258] Interestingly, although Claude3-Opus benefits the most from the background in the subproblem level evaluation, it gains less in the main problem evaluation.
> >
> > ***This is why I think it would be helpful to do tables that are {model} x {background} x {task level} x {field/science} x {background knowledge type} x {coding task type}***
> >
> > Thank you for the suggestion! Here we include an extension of table 3 with domain categories.
> > | Model                          | Math (%) | Δ Math (%) | Physics (%) | Δ Physics (%) | Chemistry (%) | Δ Chemistry (%) | Biology (%) | Δ Biology (%) | Material Science (%) | Δ Material Science (%) |
> > |---------------------------------|----------|------------|-------------|---------------|---------------|----------------|-------------|---------------|----------------------|------------------------|
> > | GPT-4-turbo-2024-04-09          | 33.33    | +20.83     | 29.33       | +4.00         | 35.14         | +21.62         | 28.00       | +12.00        | 45.45                | +16.36                 |
> > | GPT-4o                          | 25.00    | +16.67     | 34.67       | +10.00        | 32.43         | +5.40          | 24.00       | 0.00          | 47.27                | +16.36                 |
> > | Claude-3-sonnet-20240229        | 20.83    | +12.50     | 28.67       | +10.00        | 18.92         | +2.70          | 20.00       | +8.00         | 25.45                | +7.27                  |
> > | Claude-3-opus-20240229          | 29.17    | +20.84     | 23.33       | +2.00         | 35.14         | +16.22         | 24.00       | 0.00          | 29.09                | +1.82                  |
> > | Claude-3-5-sonnet-20240620      | 25.00    | +12.50     | 34.00       | +8.00         | 32.43         | +5.40          | 36.00       | +4.00         | 43.64                | +16.37                 |
> > | Gemini-1.5-pro                  | 12.50    | 0.00       | 29.33       | +8.00         | 37.84         | +18.92         | 24.00       | 0.00          | 38.18                | +9.09                  |
> > | Llama-3-70b-chat-hf             | 12.50    | 0.00       | 18.67       | +2.67         | 10.81         | +8.11          | 28.00       | +12.00        | 32.73                | +12.73                 |
> > | Mixtral-8x22B-Instruct-v0.1     | 20.83    | 0.00       | 19.33       | +2.00         | 10.81         | +2.70          | 20.00       | +8.00         | 30.91                | +7.27                  |
> > | Qwen2-72B-Instruct              | 16.67    | 0.00       | 25.33       | +5.33         | 10.81         | +2.70          | 20.00       | 0.00          | 23.64                | +7.28                  |
> > | Deepseek-coder                  | 16.67    | 0.00       | 26.67       | +5.34         | 13.51         | +10.81         | 24.00       | 0.00          | 41.82                | +7.27                  |
> >
> >
> > Additionally, we closely examine the accuracy of the Calude3-Opus main problem. Out of the four correctly answered main problems, only two remain correct when incorporating the scientists' background. The model appears to be distracted by the extra information provided and does not follow the question requirement.

---

> ### Author Rebuttal · Authors · 2024-08-31
>
> Dear Reviewers,
>
> Thank you for your constructive feedback. We have provided additional experimental results in our response. Since it is close to the end of the discussion period, please let us know if we should include anything further in the revised draft. We are more than happy to clarify if anything is unclear.
>
> Best regards,
>
> Authors

---

### Decision · Program_Chairs · 2024-09-26

**Decision:**

Accept (Poster)

**Comment:**

The reviewers found the paper to be of good quality and potentially can benefit the research community.